# The Invasive *Tradescantia zebrina* Affects Litter Decomposition, but It Does Not Change the Lignocellulolytic Fungal Community in the Atlantic Forest, Brazil

**DOI:** 10.3390/plants12112162

**Published:** 2023-05-30

**Authors:** Wagner Antonio Chiba de Castro, Giselle Cristina de Oliveira Vaz, Dalva Maria da Silva Matos, Alvaro Herrera Vale, Any Caroline Pantaleão Bueno, Luiz Fernando Grandi Fagundes, Letícia da Costa, Rafaella Costa Bonugli Santos

**Affiliations:** 1Neotropical Biodiversity Graduate Program, Federal University of Latin American Integration, Foz do Iguaçu 85866-000, PR, Brazil; 2Latin American Institute of Life and Nature Sciences, Federal University of Latin American Integration, Foz do Iguaçu 85866-000, PR, Brazil; 3Department of Hydrobiology, Federal University of São Carlos, São Carlos 13600-970, SP, Brazil

**Keywords:** nutrient cycling, microbiota, litter bags, native plants, protected areas, in situ, in vitro

## Abstract

Invasive plants affect ecosystems across various scales. In particular, they affect the quality and quantity of litter, which influences the composition of decomposing (lignocellulolytic) fungal communities. However, the relationship among the quality of invasive litter, lignocellulolytic cultivated fungal community composition, and litter decomposition rates under invasive conditions is still unknown. We evaluated whether the invasive herbaceous *Tradescantia zebrina* affects the litter decomposition in the Atlantic Forest and the lignocellulolytic cultivated fungal community composition. We placed litter bags with litter from the invader and native plants in invaded and non-invaded areas, as well as under controlled conditions. We evaluated the lignocellulolytic fungal communities by culture method and molecular identification. Litter from *T. zebrina* decomposed faster than litter from native species. However, the invasion of *T. zebrina* did not alter decomposition rates of either litter type. Although the lignocellulolytic fungal community composition changed over decomposition time, neither the invasion of *T. zebrina* nor litter type influenced lignocellulolytic fungal communities. We believe that the high plant richness in the Atlantic Forest enables a highly diversified and stable decomposing biota formed in conditions of high plant diversity. This diversified fungal community is capable of interacting with different litter types under different environmental conditions.

## 1. Introduction

Litter decomposition is the primary source of organic matter in soils [1,2] which is directly affected by species diversity [3,4] and largely mediated by microorganisms [1,5]. During decomposition, microorganisms modify the chemical composition of the litter, affecting the dynamics of carbon (C) and nitrogen (N) in the soil, which in turn affects plant growth and richness of plant communities [1,5]. Thus, through positive feedback, a decrease in plant diversity causes a reduction in the functional diversity of decomposing organisms, which reduces C and N dynamics [6,7,8].

The high amount of biomass produced by invasive plant species affects the local ecosystem by changing the primary productivity, the N dynamics, soil pH, soil microbial enzyme activity patterns [9,10], and litter decomposition rates [11]. In high-resource ecosystems, invasive plants would succeed through high rates of resource acquisition. A global meta-analysis study revealed that plant invasion may increase litter decomposition rate by 117% in invaded compared to non-invaded areas [12]. The high nutritional litter quality, due to the increase in the concentrations and flow of C and N, produced by some invasive plants [9,10] would favor certain species of fungi [11,13], which in turn increases the nutrient cycling processes [14,15,16]. Although the relationship among the litter quality, the composition of microbial community, and the decomposition rates of litter in invaded communities are still controversial [17], several studies described that invasive plants change the microbiota and consequently, the nutrient cycling [18,19,20,21]. There is also evidence that the microbial community can functionally adapt to different litter quality [22,23], leading to changes in the structure of the overall decomposing microbiota [18,21,24] and, ultimately, in the ecosystem functioning [25].

Fungi are an essential part of soil microbiota and actively promote the releasing of nutrients and organic C via litter decomposition [26] and produce lignocellulolytic enzymes, which are a type of carbohydrate-active enzyme (CAZyme). Lignocellulolytic enzymes are biocatalysts which break lignin and cellulosic materials for further hydrolysis [27], and CAZymes include enzymes that form the structure of plant biomass, such as cellulose, hemicellulose, and lignin [28]. The ability to produce these enzymes allows fungi to live in various natural conditions, being classically recognized as key organisms in nutrient cycling in forests [29,30]. Recently, there has been much interest in the changes promoted by litter produced by invasive species on fungal communities [31,32,33]. However, the effects are diverse, and the direction and magnitude of these effects are dependent on the ecosystem [34].

In this study, we evaluated whether the invasive *Tradescantia zebrina* Heynh. ex Bosse (Commelinaceae) alters plant decomposition in the Atlantic Forest and the community of decomposing fungi. We hypothesized that (1) litter from *T. zebrina* has higher decomposition rates than that from native plants, (2) litter from *T. zebrina* in areas invaded by *T. zebrina* has a higher decomposition rate than in non-invaded areas, and (3) litter from native plants in non-invaded areas has a higher decomposition rate than in invaded areas. Assuming the decrease in native regeneration because of the invasive species [35], we expect an impoverishment of decomposing organisms, altering the composition of lignocellulolytic fungi in the Atlantic Forest. Thus, we also hypothesized that (4) species composition of decomposing fungi in litter of native plants is different than that in litter of *T. zebrina* due the different litter nature, and that (5) it is mediated by the invasion. To test our hypothesis, we performed in situ and in vitro experiments.

## 2. Results

### 2.1. Decomposition of Litter under Different Environmental Conditions

The decay of different types of litter (invasive and native) under different environmental treatments (invaded and non-invaded areas), as well as under different experimental conditions (in situ and in vitro), is illustrated in Figure 1. The negative exponential model was suitable to explain the temporal decay of different litters (invasive and native) subjected to environmental treatments (invaded and non-invaded) both under in situ and in vitro experimentation (Table 1). The half-life of invasive litter was shorter than that of native litter in both experiments and environmental treatments. In the in situ experiment, the half-life of invasive litter was 59 and 53 days and that of the native litter was 126 and 134 days for invaded and non-invaded areas, respectively. In the in vitro experiment, the half-life of invasive litter was 230 and 212 days while the native litter was 315 and 317 for invaded and non-invaded areas, respectively.

Decomposition occurred faster in situ than in vitro for both environmental treatments and litter types (Figure 2). Under invasion, the rates of decomposition of invasive and native litter in situ were 1.9 and 1.7 times faster than in vitro, respectively. In the absence of invasion, the rates of decomposition of invasive and native litter in situ were 1.9 and 1.8 times faster than in vitro, respectively. There were no significant differences between the decomposition rates of the same litter types under different environmental treatments, in the different experiments over time (Table 2).

### 2.2. Community of Decomposing Fungi

To study lignocellulolytic fungi, we use selective media for fungi that have the potential to degrade lignin and fungi with cellulase activity. In the isolation process, 191 fungi were obtained (117 from day 10 and 74 from day 100), classified into 81 morphotypes (42 from day 10 and 44 from day 100). After molecular identification, all isolates were confirmed to belong to distinct morphotypes and grouped into 3 phyla and 25 different genera (16 from day 10 and 18 from day 100). We could not identify the genera of 7 morphotypes (2 from day 10 and 5 from day 100). This could be due to the absence of specimens in the databases for the region studied (ITS). Thus, to confirm the unidentified morphotype, the D1/D2 region of the ribosomal DNA 26S gene will be analyzed in future experiments. The complete composition of the fungal community for each time, environment, and litter type can be seen in Figure 3. We identified only one isolate of the phylum Basidiomycota (*Trametes* sp. 2AQ) in the invaded area on day 10 from the native substrate (litter bag). The low representation of this phylum is probably because basidiomycetes are difficult to isolate from soil because they characteristically grow more slowly in culture than ascomycetes. We identified 2 morphotypes from the phylum Mucoromycota, predominantly in day 10 when it was identified in 8 of the 10 areas, 5 invaded and 3 non-invaded areas (*Mucor* sp. 1D and *Mucor* sp. 2I). The remaining 78 morphotypes were from the phylum Ascomycota (96%). The most abundant genus was *Fusarium* (25%), followed by *Aspergillus* (12%), *Colletotrichum* (7.4%), *Penicillium* (6.2%), and *Trichoderma* (6.2%).

*Fusarium* sp. 2J and *Mucor* sp. 1D were highly prevalent on day 10, but were not observed on day 100. *Penicillium* sp. 2G was isolated from all 10 areas on day 10 and from most areas on day 100. On day 10, 52.2% of the morphotypes observed from the invaded areas were isolated from only one of the five areas. On the same day, 63.0% of the morphotypes observed from the non-invaded areas were isolated from only one of the five areas. On day 100, 56.7% of the morphotypes from the invaded areas were isolated from only one of the five areas. On the same day, 50% of the morphotypes observed in the invaded areas were isolated from only one of the five areas.

The distribution of total morphotypes isolates was 54% for the invaded area and 46% for the native area. Evaluating separately each area and each litter studied, in the invaded area 25% of morphotypes were isolated from the invasive litter and 29% from native litter and in the native area 23% were isolated from the invasive litter and 23% from native litter. There were no significant differences in the composition of the lignocellulolytic fungal community between the invaded and non-invaded areas both for day 10 (F = 0.77; *p* = 0.39) and day 100 (F = 0.69; *p* = 0.42). Neither the type of litter nor interactions between litter types and environment explained the lignocellulolytic fungal community composition (Table 3).

## 3. Discussion

Our results demonstrate that litter from *T. zebrina* decomposes more quickly than litter from native plants of the Atlantic Forest. However, the invasion of *T. zebrina* does not influence either the decomposition rates of litter or the composition of lignocellulolytic fungi.

The higher decomposition rate of *T. zebrina* than that of native litter, in both invaded and non-invaded areas, may be associated with the quality of the litter. This is probably due to a higher concentration of N commonly observed in invasive plants than in native species of invaded communities [11,36]. For example, the litter of two invasive species, *Buddleja asiatica* and *Myrica faya*, had higher amount of N and decomposed faster than litter from natives in invaded areas in Hawaii [37]. The accumulation of N and other nutrients in the tissues of *Tradescantia fluminensis* Vell. (Commelinaceae), a morphologically similar species of the same genus and life-form of *T. zebrina*, accelerates the decomposition of its litter and, together with the large biomass of *T. fluminensis*, alters the availability of nutrients in invaded forest remnants from New Zealand [38,39].

The lability of *T. zebrina* litter could also explain its faster decomposition rates. Similar to *T. fluminensis* [39], *T. zebrina* apparently has lower levels of lignin and higher levels of labile compounds than native litter in the Atlantic Forest. During the beginning of the decomposition process, a high N content can support large microbial populations that quickly consume labile compounds, which results in a faster loss of mass (1), corroborating our decay results. The high concentration of N, both due to the quality and quantity of the litter, can create positive feedback that promotes the proliferation of the invasive species and/or other exotic species [12,40]. Invasive species, such as *T. zebrina*, with faster decomposition rates than native ones, can benefit from the high nutritional support provided by the invasion and, in certain situations, outcompete the native species [41].

The lower rate of decomposition found for native litter could be due to the life-form diversity of native species. In our experiment, the invasive litter was of exclusively herbaceous origin, while the native litter came from plants with different life-forms and included trees and branches fragments. In general, the leaves of herbaceous species are more easily decomposed than the leaves of arboreal species [42], since leaves are mainly composed of soluble carbohydrates, which decompose quickly [43]. Decomposition rates can also vary with plant tissue type [44]. For example, the leaves of *Lythrumsalicaria*, invasive in wetlands in the northern United States, decompose faster than native species (*Typha* sp.) in invaded areas, while the invader’s stems decompose more slowly [45]. Even litter composition, whether it is composed of one species or a mixture of species, can influence decomposition rates [7,46]. Therefore, the composition of the native litter, which included various plant sources with different life-forms and/or tissues, may have resulted in the lower decomposition rates. However, our native litterbags made of a diversity of species and life-forms from the non-invaded areas are a good approximation of the dynamics of litter decomposition in natural habitats.

In general, in situ decomposition was faster than in vitro decomposition. The wear and tear in the natural environment by weathering; the action of more organisms, such as shredders; and loss by sedimentation of particles smaller than the mesh of the litter bags, may favor higher decomposition rates in in situ experiments [47]. Despite the absence of these factors and hence less realistic results in in vitro experiments [48], investigations based on laboratory experiments are important tools for establishing causal links between selected variables and chemical or biological responses [35,47]. In the in vitro experiment, the contribution of microbiota and chemical oxidation to the decomposition process can be measured [49]. Chemical oxidation in aerobic incubations is responsible for 1 to 5% of all oxygen consumption from the decomposition of different plants [50]. Thus, by subtracting the effect of chemical oxidation (1 to 5%), we can estimate the microbial contribution to the general decomposition process. Our results suggest that the microbiota is responsible for 47–56% of all litter degradation. This highlights the importance of studies of the role of microbiota in nutrient cycling processes in biological invasions [35].

In temperate forests, the litter decomposition rate is often different in invaded areas compared with that in non-invaded areas [18,19,20]. However, our results do not corroborate the current literature. We believe that the lack of differences observed in our study can be attributed to the environmental characteristics of the Atlantic Forest, including the diversity of lignocellulolytic fungi. The climate [51], the diversity of decomposing organisms, and the very nature of a highly diversified plant litter [52], are determining factors in the rate of decomposition [53,54], and regulate the activity of decomposers of organic matter in the soil [55]. The litter heterogeneity of the Atlantic Forest provides a greater diversity of niches for the community of decomposers, keeping the fungal diversity more stable, even under forest gradients [56]. These characteristics favor a highly diversified and active decomposing microbial community [57]. The natural range of different litter from the Atlantic Forest provides an opportunity for this decomposing community, formed over a long time in conditions of high plant diversity, which consequently has high taxonomic and functional diversity, to optimise the cycling of nutrients [44,58], including those from invasive litter. The absence of a pattern in the composition of the lignocellulolytic fungal community corroborates this argument.

Despite the changes in the quality of litter due to the invasion, the diversity of lignocellulolytic fungi remained high, which likely contributed to sustaining the ecological processes related to plant decomposition. Even when litter is difficult to decompose for part of the microbiota, interactions between decomposers can improve degradation efficiency [58]. In the context of the high diversity of microbiota existing in the Atlantic Forest [57], it is likely that these interactions enhance the efficiency of nutrient cycling of many types of litter, even under different environments and coevolutionary histories between litter and decomposing microbiota [44]. Thus, the use of different substrates by different fungal species promotes the maintenance of the diversity of soil fungi [59]. Furthermore, fungi exhibit a variety of morphological and adaptive traits under different forest microclimates [60]. These factors can partially explain the absence of a difference in the decomposition rates of the same litter type under invaded and non-invaded areas. That is, the decomposition conditions were likely optimal, even under the different environments.

Our study sought to isolate fungi with lignocellulolytic potential using the reagent guaiacol and the addition of CMC (carboxymethyl cellulose) as the only C source. Standard isolation makes it impossible to recover microorganisms that occur in low numbers which are responsible for critical ecosystem services [61]. We added the reagent guaiacol to the culture medium to select fungi that produce ligninolytic enzymes [62]. For growth in the CMC culture medium, fungi must produce certain enzymes, especially endoglucanases, enzymes of the cellulolytic complex responsible for initiating the hydrolysis of cellulose [63]. These enzymes, known as CAZymes, are classified into several hundred different enzyme protein families. However, a CAZyme family often houses proteins from a broad taxonomic range, covering different taxonomic classes and even different kingdoms [64]. In this way, we believe that the applied methodology was adequate to contemplate the three phyla identified in our study.

Most of the fungal morphotypes identified in our study belong to the phylum Ascomycota. This phylum, which is dominant in the soil, can use a higher number of resources than others and may be better equipped to withstand environmental disturbances [65]. This versatility may explain the significant occurrence of the phylum Ascomycota in the plant litter of the Atlantic Forest, especially its dominant presence in litter from the invaded area, since the litter of *T. zebrina* has allelopathic properties [66,67]. The most abundant genus was *Fusarium*, a genus widely represented among the filamentous fungi, associated with soil and plants worldwide. It has many symbiotic relationships as a phytopathogen, synthesising CAZymes to obtain energy via plant litter [68]. Analysis of the gene expression profile of the fungus *Fusarium graminearum*, for example, revealed that most of the CAZyme genes related to cell wall degradation are up-regulated during plant infection [69]. Owing to its wide occurrence and enzymatic efficiency, *Fusarium* species may be primarily responsible for the degradation of plant litter in our experiment.

Our results show that the time since the beginning of the experiment influences the composition of lignocellulolytic fungi, as *Alternaria* sp.; *Bipolaris* sp.; *Epicoccum* sp.; *Montagnula* sp. *Trametes* sp.; *Volutella* sp.; *Wiesneriomyces* sp.; and the isolate from the taxonomic group Nectriaceae were exclusively found on day 10. In contrast, *Eutypella* sp.; *Cladosporium* sp.; *Lasiodiplodia* sp.; *Myrothecium* sp.; *Nigrospora* sp., *Pestalotiopsis* sp., *Sarocladium* sp., and the isolates belonging to the taxonomic levels Pleosporales and Bionectriaceae were exclusively found on day 100. This profile shows the rapid ecological succession of the fungal community that decomposes plant litter [70], with highly active morphotypes in the initial (labile/soluble) and final (recalcitrant) phases of decomposition [71]. However, for both invaded and non-invaded areas, regardless of the sampling time, we found that at most 50% of morphotypes were associated with more than a single sampling area, even considering sampling areas within the same environmental treatment. This reinforces the high diversity of lignocellulolytic fungi in the Atlantic Forest, where different microorganisms may be naturally available to decompose various litter types [72,73]. The high diversity of the lignocellulolytic cultivated fungal community indicates that the performance of a specific fungal morphotype under plant litter is mediated by opportunity. Thus, the chances of the same fungus morphotype occurring in plant litter from several microhabitats is low. This could explain the absence of an effect of invasion and litter quality on the composition of the lignocellulolytic cultivated fungal community.

## 4. Materials and Methods

### 4.1. Description of the Study Area

We conducted this study at the Iguaçu National Park (PNI) in southern Brazil (25°05′ and 25°41′ S; 53°40′ and 54°38′ W). The PNI is the largest protected area in the Atlantic Forest in Brazil. It is 185,262.5 ha [74] and is part of one of the most important biological areas in South-Central South America, which has more than 600 thousand hectares of protected areas and another 400 thousand hectares of primitive forests [75]. The region has a subtropical climate with temperatures between 15 °C and 26 °C, and an average annual rainfall of 1841 mm [76]. We selected ten areas for the experiment, five of which were invaded by *T. zebrina* (60 to 90% coverage of *T. zebrina*, with no other dominant species) and five non-invaded (without *T. zebrina*).

### 4.2. T. zebrine

The herbaceous *T. zebrina* is originally from Mexico and northern Central American countries and was brought to Brazil for ornamental purposes [77]. It has a high capacity for adaptation to different Brazilian biomes, with records of invasion in the Cerrado [78] and Atlantic Forest [77,78]. It is an herbaceous plant, growing up to 25 cm with purple-green glabrous leaves, which quickly multiply by stem fragments, and is also considered a weed in agriculture [79]. It is considered a strong competitor, affecting the diversity of species in forest fragments [77], mainly in the forest edge regions, inhibiting tree regeneration [80]. It is cited in several lists of invasive species [81,82,83]. Laboratory tests have revealed this plant is potentially allelopathic [84], and some studies attribute part of its dominance in areas invaded by its large biomass and allelopathy [77,85,86]. In the PNI, the density of *T. zebrina* negatively affects the height of the regenerants, impairing the recruitment and development of the plant community [80]. Additionally, invaded communities have higher predation rates, which increases the indirect effects of the invasion on the development of native species [87].

### 4.3. Litter Bag Preparation

To compare the decomposition rate of the litter from invaded and non-invaded areas, we collected live fragments of (1) *T. zebrina* from five invaded areas and (2) native trees/shrubs/herbs from the five non-invaded areas. The invasive and native fragments were dried separately in an oven at 55 ± 5 °C until they attained a constant weight. We prepared large litter bags (15 × 15 cm) for an in situ experiment and small litter bags (5 × 5 cm) for an in vitro experiment, containing 5 g and 2 g litter per bag, respectively. For each size bag, two types of litter were used, one with only dry fragments of the invader (invasive litter bags) and the other with dry fragments of native species (native litter bags). The composition of species and types of fragments in the native litter bags was random. Both litter bags were made using a plastic mesh of 5 mm pore diameter to allow access to macroinvertebrates.

### 4.4. In Situ Experiment

We started the experiment in November 2018. In each of the 10 areas of the experiment (5 invaded areas and 5 non-invaded areas), we deposited 60 large litter bags, 30 invasive and 30 native litter bags. The litter bags were deposited alternately, approximately 30 cm from each other. They were placed on the soil surface without removing the litter layer and tied to tree trunks with nylon thread to avoid displacement due to animals. Ten sampling days were established over an eight-month period (0, 10, 20, 30, 50, 70, 100, 140, 180, and 240 days from the beginning of the experiment). On each sampling day, we collected six litter bags in each area, three of each type (sub-replicates).

### 4.5. In Vitro Experiment

For this controlled experiment we collected soil from the A horizon in invaded and non-invaded areas, and litter from non-invaded areas. In four 0.15 m^2^ (0.5 × 0.3 m) trays, under controlled conditions of temperature (25 ± 3 °C), light (3000 lux) and photoperiod (12/12 h), we simulated the environmental conditions of invaded and natural forest (presence and absence of *T. zebrina*, respectively). We prepared 60 invasive and 60 native litter bags. In 60 trays, we spread soil and litter from the invaded areas and in each of 30 trays we placed 1 invasive litter bag and in the other 30 we placed native litter bags. In other 60 trays we spread soil and litter of the non-invaded areas and in each of 30 trays we placed 1 invasive litter bag while in the other 30 we spread 1 native litter bag. We collected three litter bags from each tray on the same sample days as the in situ experiment.

### 4.6. Litterbags Processing

Each collected litterbag was placed separately in a plastic bag. In the laboratory, we disposed each litterbag on an individual sheet of paper, to avoid the loss of plant fragments in the drying process. Litterbags were dried at 55 ± 5 °C for 72 h. The remaining particulate plant material was measured gravimetrically.

### 4.7. Isolation and Identification of Decomposing Fungi from Plant Litter

To analyze the richness of lignocellulolytic fungi, we deposited an additional 12 large litter bags (15 × 15 cm), 6 with invasive and 6 native litter, in each of the 10 areas. We sampled litter bags twice in four months, exploring two stages of decomposition: (1) decomposition of labile/soluble material (day 10) and (2) decomposition of recalcitrant material (day 100). At each sampling time, we collected six litter bags from each area, three of each type of litter. We mixed the content from the bags with the same type of litter collected in each area to prepare a composite sample. Thus, we obtained 40 composite samples, 20 from day 10 and 20 from day 100. That is, at each sampling time, we obtained five samples of each treatment. In the laboratory, the 40 samples were cleaned by washing with distilled water to remove soil and other organisms present [88]. Then, the samples were processed using the particle filtration method [89]. This process consisted of homogenisation using a high-speed industrial blender for two minutes in 200 mL of sterile distilled water, after which the particulate material was washed with sterile distilled water jets (2 washes with 100 mL) in stainless steel sieves (600 µm, 300 µm, 150 µm, then 75 µm mesh size). The particles retained in the smallest mesh sieve were used to perform three serial dilutions (10:1, 10:2, and 10:3). Then, we plated 0.1 mL of the 10:3 dilution [90,91]. For each plating, we performed triplicates with 2% malt extract supplemented with guaiacol (20 g/L of malt extract, 4 mM guaiacol, and 15 g/L of bacteriological agar) according to Kiiskinen et al. [92] and CMC 1% (10 g/L of carboxymethylcellulose, 6.7 g/L of yeast extract, and 15 g/L of bacteriological agar) according to Makhuvele et al. [93]. All culture media were prepared with 10 mg/L of chloramphenicol to inhibit bacterial growth. We incubated the plates at 28 °C, and fungal growth followed for 30 days [94]. The isolates were purified and preserved by the Castellani method [95], and cryopreservation with 10% glycerol and kept at −80 °C. We stored the preserved material in the Culture Collection of Microorganisms of Biotechnological and Environmental Importance (UNILA CCMIBA). Purification consisted of raising each isolate separately until confirmation that the isolate represented an individual with unique morphological characteristics (until there was a single morphotype). The use of the concept of morphotype has been useful in estimating the number of fungi since the species is conventionally the basic unit in biodiversity studies [96].

We analyzed the purified fungi for the determination of morphotype (morphological units). We identified the different morphotypes morphologically by microscopy of the mycelium. We used a stereoscopic magnifying glass and microscopically prepared slides and microcultures colored with blue cotton lactophenol. Three replicates of each of the isolates were inoculated at the same time and incubated for 4 weeks. The growth rate was determined by measuring the average diameter of the colony (cm). Other colony characteristics, including color (above and reverse), elevation, texture, type of mycelium, margin shape, density, zonale, and effects of the fungi on the medium, were examined. The morphology of fungi was observed under a compound microscope to examine the shapes forming from the arrangement of spores [97]. For unsporulated fungi, each isolate was characterized as a distinct morphotype. We photo-documented the images for deposit at the CCMIBA and morphotype archives.

We selected 1 representative of each morphotype, 102 isolates, for DNA extraction and molecular identification. We extracted the genomic DNA of the isolates according to the protocol described by Raeder and Broda [98]. We amplified the ITS1-5.8S-ITS2 region of the ribosomal DNA with the universal primers for fungi, ITS-1 (5′TCCGTAGGTGAACCTGCGG-3′) and ITS-4 (5′TCCTCCGCTTATTGATATGC-3′), which amplify a region of approximately 600 bp [99]. We prepared the reactions in a final volume of 25 μL, containing genomic DNA (1.0 ng/μL), 2.5 μL of enzyme buffer, 0.5 μL of dNTPs (10 mM), 0.6 μL of the ITS-1 primer (10 µM), 0.6 μL of the ITS-4 primer (10 µM), and 0.3 μL of Taq DNA polymerase. We programmed the thermal cycler for an initial denaturation at 94 °C for 2 min; followed by 30 cycles at 94 °C for 1 min, 55 °C for 1 min, and 72 °C for 1 min; and a final extension at 72 °C for 10 min. We analyzed the PCR product on agarose gel (1% *w*/*v*) with GelRed^®^ Nucleic Acid Gel Stain (Biotium, Fremont, CA, USA), with the addition of a molecular marker (1 Kb Plus DNA Ladder). Then, we purified the product using the enzyme ExoSap-IT. The samples were then sequenced using the SANGER method at ACTgeneAnalyzesMoleculares^®^ (Alvorada, Rio Grande do Sul state, Brazil).

The sequences obtained were blasted against the National Center for Biotechnology Information (Genbank-NCBI) database and the MycoBank database (Appendix A). We performed the sequence alignment using the online program Clustal W and phylogenetic and molecular analyses were conducted using MEGA software version 7.0 [100]. We used the Kimura 2-parametermodel [101] to estimate the evolutionary distance and the Maximum Likelihood (ML) algorithm for phylogenetic reconstructions, with the bootstrap value calculated from 1000 replicates. After analysing similarity using Blast N, we grouped the isolates classified at the genus level. We evaluated the isolates belonging to the same genus through phylogenetic analysis. To confirm the previously defined morphotype, we then compared the fungi grouped in the same branch using the global alignment of the Needleman–Wunsch Global sequence.

### 4.8. Analysis of the Community of Fungal Decomposers

We estimated the rate of decomposition using a first-order negative exponential decay model [102] widely used to adequately capture the decay dynamics in many systems [42,72]:(1)$$X_{t}=X_{0}e^{-\mathrm{kt}}$$
where X_t_ and X_0_ correspond to the initial and final mass of the waste, respectively, t is the time (days), and k is the mass loss constant.

We calculated the half-life (t_½_) of litter using the equation:(2)$$t_{\text{½}}=\frac{\ln\left(0.5\right)}{-k}$$
where k is a constant of mass loss in each litter bag of each experiment.

We used linear regression to compare the decomposition rates between in situ and in vitro experiments for both types of litter (invasive and native). The line slope (regression parameter t) represents the rate between decomposition speeds of the two methodologies. To verify significant differences between the decay of litter types in the different environmental treatments (invaded and non-invaded), for both decomposition experiments, we used a generalized linear mixed model (GLMM) using the ‘glmer’ function in the ‘lme4’ package [103] with a binomial distribution and temporal pseudo-replication. The pseudo-replicates were sets of samples carried out over time. Thus, the fixed effect included sampling time as a categorical factor, with 10 levels (each re-sampling expressed in the estimation of the mass of the litter bag), as well as its interaction with the effects of (1) the type of litter bag and (2) environmental treatment. The structure of random effects includes interception for treatments and temporal sampling [104].

### 4.9. Data Analysis

We used morphotype data to build distance matrices with Jaccard presence/absence data (0 < dJ < 1, with 0 being identical and 1 being completely different; [105]). To verify significant differences in the composition of the morphotypes between the invaded and non-invaded areas at the different stages of decomposition (day 10 and day 100), we determined the beta diversity [106] using the ‘betadisper’ function in the ‘betapart’ package [107]. To check if there were significant differences regarding dissimilarity (1) between the areas (invaded and non-invaded) within the spatial replicates, and (2) between the different types of litter within the sample replicates, in the different stages of decomposition (day 10 and day 100), we conducted a three-way PERMANOVA [108] using the ‘adonis’ function of the ‘vegan’ package [109], where ‘time’, ‘area’, and ‘litter’ were the three factors. We performed all statistical analyses in the R statistical environment [110].

## 5. Conclusions

Our study allowed a useful estimate of invasive and native litter decomposition in in situ and in vitro methods under different environmental conditions. The fastest decomposition of *T. zebrina* was probably due to the litter quality. Unlike the current literature, plant decomposition does not change in invaded compared with non-invaded environments. Likewise, *T. zebrina* does not encourage a specialized fungal decomposing community. The high plant diversity in the Atlantic Forest probably supports a fungal community that preserves the historical experience of a wide variety of litter, and easily finds ways to utilize different substrates, including exotic ones.

## Figures and Tables

**Figure 1 plants-12-02162-f001:**
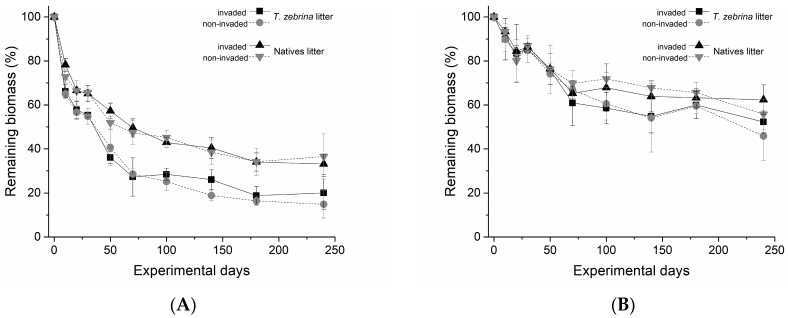
Temporal variation of the decomposition of *T. zebrina* and native species in situ (**A**) and in vitro (**B**) in Atlantic Forest areas/treatments invaded and non-invaded by *T. zebrina* throughout the experiment.

**Figure 2 plants-12-02162-f002:**
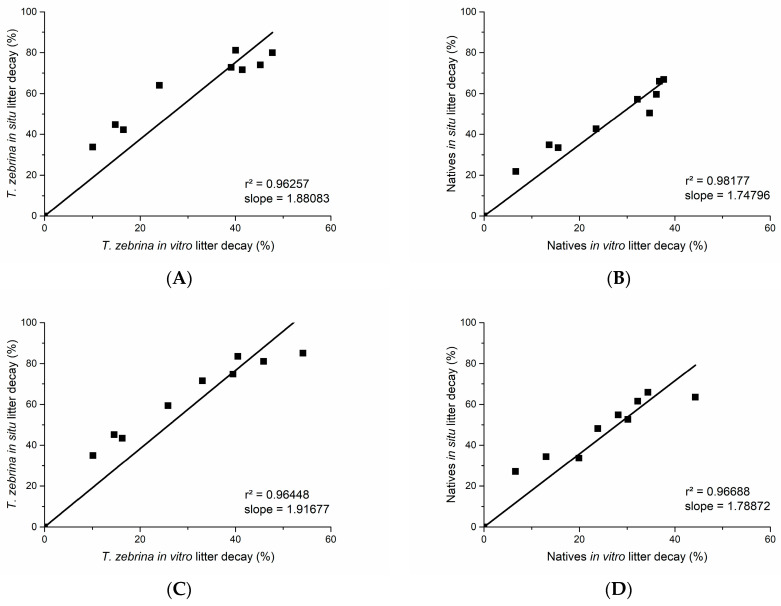
Linear regression between the decomposition values of litter in in situ and in vitro experiments of *T. zebrina* and native species. (**A**,**B**) represent invaded areas, and (**C**,**D**) represent non-invaded areas. The model adjustment values (r^2^) and slopes of the lines are shown.

**Figure 3 plants-12-02162-f003:**
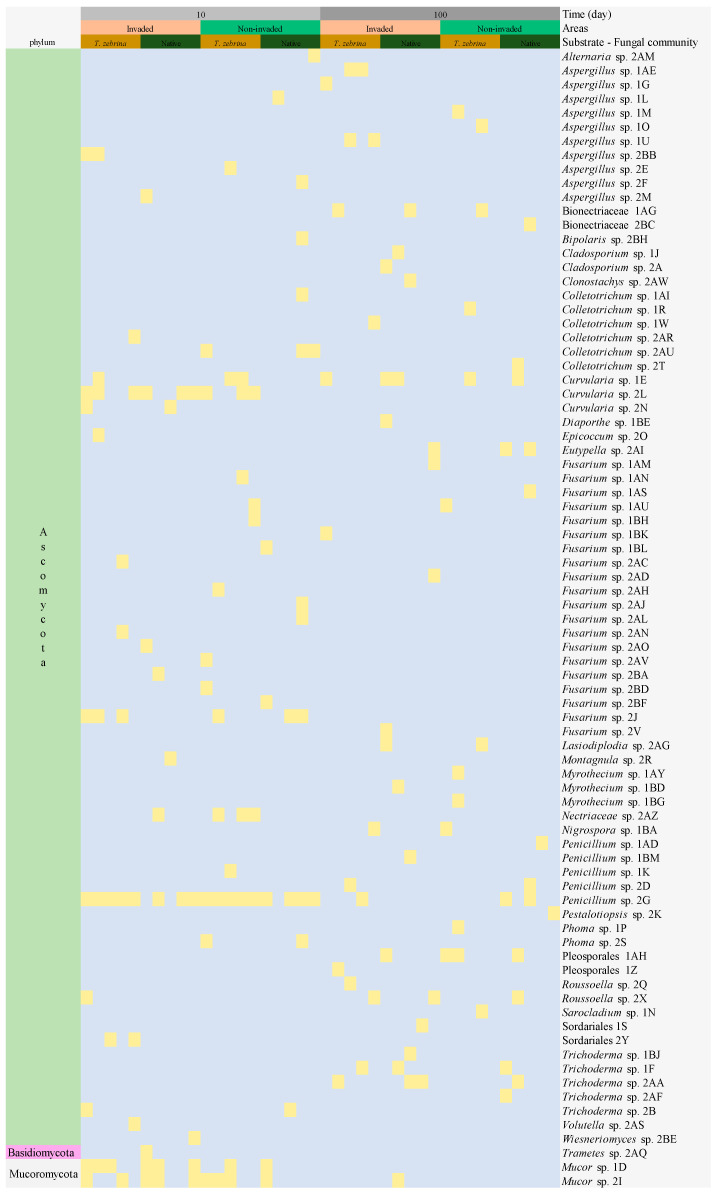
The entire composition of the fungal community at experimental timepoints (10 and 100 days), under different treatments (invaded and non-invaded areas), and on each substrate (*T. zebrina* and native species). Yellow squares represent the fungi presence.

**Table 1 plants-12-02162-t001:** Results of the first-order negative exponential decay model (Olson, 1963) and half-life (t_½_) for different experiments (in situ and in vitro), litter type (*T. zebrina* and native species), and environmental treatments (invaded and non-invaded).

Experiment	Litter Bags	Treatment	r^2^	k (Days^−1^)	k (SD)	*t* _½_
In situ	*T. zebrine*	invaded	0.81	0.01165	0.00251	59
		non-invaded	0.89	0.01316	0.00227	53
	Natives	invaded	0.83	0.00550	0.00096	126
		non-invaded	0.76	0.00519	0.00110	134
In vitro	*T. zebrine*	invaded	0.79	0.00302	0.00055	230
		non-invaded	0.89	0.00327	0.00042	212
	Natives	invaded	0.73	0.00220	0.00045	315
		non-invaded	0.81	0.00219	0.00037	317

r^2^ = adjustment factor of the decay model; k = decay coefficient.

**Table 2 plants-12-02162-t002:** Results of the mixed linear model adjusted for differences in litter decay values between environmental treatments (invaded and non-invaded) in both in situ and in vitro experiments. We used the decay values of *T. zebrina* and native litter over time.

Response Variable	Factor	Estimate	SE	DF	T Value	*p* Value
*T. zebrina* litter	In situ	−0.01647	0.00986	80	−1.670	0.099
	In vitro	−0.001068	0.019613	44	−0.054	0.957
Native litter	In situ	−0.009728	0.009725	80	−1.00	0.320
	In vitro	0.004726	0.015915	44	0.297	0.768

**Table 3 plants-12-02162-t003:** Multivariate permutation analysis on the effect of decomposition time (10 and 100 days), type of litter (native and invasive), and environmental treatment (invaded and non-invaded by *T. zebrina*) and of the interactions between effects on the composition of the Atlantic Forest lignocellulolytic fungal community, based on Jaccard distance matrices (presence/absence data).

Factor	df	Sum of Squares	Mean Squares	F Value	*p* Value
Time	1	1.756	1.756	4.313	***
Environment	1	0.417	0.417	0.024	0.388
Litter	1	0.382	0.382	0.022	0.563
Time × Environment	1	0.3871	0.38707	0.9509	0.53095
Time × Litter	1	0.4101	0.41009	1.0075	0.41256
Environment × Litter	1	0.5309	0.53094	1.3044	0.08549
Time × Environment × Litter	1	0.4363	0.43630	1.0719	0.30807
Residuals	32	13.0254	0.40704		

*** < 0.0001. Analysis conducted using PERMANOVA.

## Data Availability

Data used in this study can be available by contacting wagner.castro@unila.edu.br.

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
