# Peer review of "The Invasive Tradescantia zebrina Affects Litter Decomposition, but It Does Not Change the Lignocellulolytic Fungal Community in the Atlantic Forest, Brazil"

_plants, 2023, doi:10.3390/plants12112162_

Round 1
Reviewer 1 Report
The general meaning of the manuscript, which is presented for publication, consists in assessing the rate of decomposition of plant material belonging to invasive specie Tradescantia zebrina and local plant material in the areas of invasion and in areas free from invasive species. The experiments were carried out both in natural conditions and in laboratory conditions. The main results are to demonstrate that under all experimental conditions, plant material belonging to an invasive species decomposes at a faster rate under all experimental conditions. At the same time, the composition of the fungal community studied by the authors does not undergo significant changes.
The study, of course, is of some interest, but there are several fundamentally important problems in the work that should be noted.
First, the authors focused on the fact that the main feature of plant material is its belonging to an invasive species. Obviously, the problem of invasiveness may not be significant. Since the authors themselves point out (lines 181-193) that the structure of tradescantia biomass, consisting mainly of leaves, a higher nitrogen content, a low proportion of lignin components, etc., but not the invasive nature of plant material, can become the cause of high decomposition rates. And this is a good idea, since all these parameters can be estimated numerically without much difficulty, but the authors of these analyzes did not perform. Another problem is that out of the whole variety of microorganisms involved in the decomposition of plant residues, the authors paid attention only to a small part of cultivated fungi, although today it is obvious that there are much more real participants in decomposition and a very large part of them are prokaryotes. I do not at all reproach the authors for not using modern methods of high throughput sequencing, which are quite available today. But the choice of analysis method should have some justification in the introduction.
It is possible that after a significant revision of the article, it could be published in the journal Plants, but I leave the final decision to the editor, since I am not quite sure that the work is up to the level of the journal.
However, in case the manuscript is recommended for revision, I have left notes in the text of the article. Here is one of my comments. Figure 3 is very difficult to read due to poor layout. Designations of the area, substrate, time should be moved to the top of the figure and made more clear. The use of the same color scheme to designate area and substrate misleads the reader. In addition, for a clearer understanding of the patterns of distribution of fungal species in the variants of the experiment, I would advise you to either perform a cluster analysis of samples, or use some kind of ordination (like PCI).

Author Response
The general meaning of the manuscript, which is presented for publication, consists in assessing the rate of decomposition of plant material belonging to invasive specie Tradescantia zebrina and local plant material in the areas of invasion and in areas free from invasive species. The experiments were carried out both in natural conditions and in laboratory conditions. The main results are to demonstrate that under all experimental conditions, plant material belonging to an invasive species decomposes at a faster rate under all experimental conditions. At the same time, the composition of the fungal community studied by the authors does not undergo significant changes.
The study, of course, is of some interest, but there are several fundamentally important problems in the work that should be noted.
- We would like to thank you for the comments. The suggestions provided are precious and we expect to explore all of them. We believe our new manuscript version is quite improved.
First, the authors focused on the fact that the main feature of plant material is its belonging to an invasive species. Obviously, the problem of invasiveness may not be significant. Since the authors themselves point out (lines 181-193) that the structure of tradescantia biomass, consisting mainly of leaves, a higher nitrogen content, a low proportion of lignin components, etc., but not the invasive nature of plant material, can become the cause of high decomposition rates.
- We thank the comment. The reviewer is right. The nature of the invasive detritus is consistently different from the native detritus. For this reason, we expected different fungi communities according to the different detritus nature. We were surprised because it didn´t happen.
And this is a good idea, since all these parameters can be estimated numerically without much difficulty, but the authors of these analyzes did not perform. Another problem is that out of the whole variety of microorganisms involved in the decomposition of plant residues, the authors paid attention only to a small part of cultivated fungi, although today it is obvious that there are much more real participants in decomposition and a very large part of them are prokaryotes. I do not at all reproach the authors for not using modern methods of high throughput sequencing, which are quite available today. But the choice of analysis method should have some justification in the introduction.
- We thank the comment. Unfortunately, we didn´t have resources to explore high throughput sequencing, considering the number of treatments and replicates in this study. We explored 10 areas, each one with 60 litterbags (a total of 600 litterbags), as well as plus 120 litterbags under lab condition. So, we choose to explore a cultivated method to valorize the spatial and temporal replications in several treatments.
It is possible that after a significant revision of the article, it could be published in the journal Plants, but I leave the final decision to the editor, since I am not quite sure that the work is up to the level of the journal.
However, in case the manuscript is recommended for revision, I have left notes in the text of the article. Here is one of my comments. Figure 3 is very difficult to read due to poor layout. Designations of the area, substrate, time should be moved to the top of the figure and made more clear. The use of the same color scheme to designate area and substrate misleads the reader. In addition, for a clearer understanding of the patterns of distribution of fungal species in the variants of the experiment, I would advise you to either perform a cluster analysis of samples, or use some kind of ordination (like PCI).
- We thank the comments. We detailed our responses below.
- Line 42-44) “Is would be necessary to clariy what exactly the differences are.
Sounds strange. Invaiders have distinct requirements but produce the high amount os biomass.”
- We thank the comment. We included We reformulated our Introduction and we believe this new version is improved.
- Line 47-48) “It sounds unespected and requires at least a little explanation and not just a reference to a specific publication.”
- We thank the comment. We included the phrase “In high-resource ecosystems, invasive plants would succeed through high rates of resource acquisition”.
- Line 50) “why is there no positive feedback for locals? Extract please an explanation from the reference cited.”
We suppressed the following part: “in invaded ecosystems, resulting in positive feedback for their own development. In addition, the high nutritional quality litter”, and modified the paragraph:
“The high amount of biomass produced by invasive plant species affects the local ecosystem by changing the primary productivity, the N dynamics, soil pH, soil microbial enzyme activity patterns [12,13], litter decomposition rates [14]. In high-resource ecosystems invasive plants would succeed through high rates of resource acquisition. A global meta-analysis study revealed that plant invasion may increase litter decomposition rate by 117% in invaded compared to non-invaded areas [11). The high nutritional litter quality, due to the increase on the concentrations and flow of C and N, produced by some invasive plants (12, 13) would favor certain species of fungi (10,14), which in turn increases the nutrient cycling processes (9,14,15,16). Although the relationship among the litter quality, the composition of microbial community, and the decomposition rates of litter in invaded communities are still controversial (17).”
We change the paragraphs as follows:
“Several studies described that invasive plants change the microbiota and consequently, the nutrient cycling [18-21]. There are also evidences that the microbial community can functionally adapt to different litter quality [22,23] leading to changes on the structure of the overall decomposing microbiota [18,21,25] and, ultimately, in the ecosystem functioning [26].
Fungi are an essential part of soil microbiota and actively promote the releasing of nutrients and organic C via litter decomposition [27] and, produce lignocellulolytic enzymes, which are a type of carbohydrate-active enzyme (CAZyme). Lignocellulolytic enzymes are biocatalysts which break lignin and cellulosic materials for further hydrolysis [28], and CAZymes include enzymes that form the structure of plant biomass, such as cellulose, hemicellulose, and lignin [29]. The ability to produce these enzymes allows fungi to live in various natural conditions, being classically recognized as key organisms in nutrient cycling in forests [30,31]. Recently, there has been much interest in the changes promoted by litter produced by invasive species on fungal communities [32-34]. However, the effects are diverse, and the direction and magnitude of these effects are dependent on the ecosystem [35].”
We suppressed the following part: “Thus, plant invasions can profoundly impact the quality and quantity of litter and the dependence between producers and decomposers [21]. These impacts result in changes not only in the plant community but also in…”
- Line 51) “Is high nutritional quality characteristic of any invaders?”
We thank the comment. Answering your question, no. We change the paragraph as follow “The high nutritional litter quality, due to the increase on the concentrations and flow of C and N…” and we suppressed the following part: “in invaded ecosystems, resulting in positive feedback for their own development (12). In addition, the high nutritional quality litter…”
- Line 77 – 81) “Could you state the source of this hypothesis? From the introduction have a higher decomp rate. For example, one of the possible sources of this hypothesis is associated with the selection of an invader-specific decomposition microbiota. The same 2.1: But native-specific microbiota.”
We thank the comment. In the Introduction section, we explain this hypothesis “The high nutritional litter quality, due to the increase on the concentrations and flow of C and N, produced by some invasive plants (12, 13) would favor certain species of fungi (10,14), which in turn increases the nutrient cycling processes (9,14,15,16).”
- Line 84) “It is too simple, try to formulate a hypothesis that would explain why these communities are different.”
We thank the comment. We change the hypothesis as follow: “(3) species composition of decomposing fungi in litter of native plants is different than that in litter of T. zebrina due the different litter nature”
- Table 1) “I it seems to me that this table should be done in a different way. I just didn’t understand anything about it.”
We thank the contribution. We suppressed the Table 1.
- Line 125) “Cultures?”
We thank the comment. We change the paragraph as follow:
“To study lignocellulolytic fungi we use selective media for fungi that have the potential to degrade lignin and fungi with cellulase activity. In the isolation process, 191 fungi were obtained (117 from day 10 and 74 from day 100), classified into 81 morphotypes (42 from day 10 and 44 from day 100). After molecular identification, all isolates were confirmed to belong to distinct morphotypes and grouped into three phyla and 25 different genera (16 from day 10 and 18 from day 100). We could notidentify the genera of seven morphotypes (2 from day 10 and 5 from day 100). This could be due to the absence of specimens in the databases for the region studied (ITS). Thus, to confirm the unidentified morphotype, the D1/D2 region of the ribosomal DNA 26S gene will be analyzed in future experiments. The complete composition of the fungal community for each time, environment, and litter type can be seen in Figure 3. We identified only one isolate of the phylum Basidiomycota (Trametes sp. 2AQ) in the invaded area on day 10 from the native substrate (litter bag). The low representation of this phylum is probably because basidiomycetes are difficult to isolate from soil because they characteristically grow more slowly in culture than ascomycetes. We identified two morphotypes from the phylum Mucoromycota, predominantly in day 10 when it was identified in 8 of the 10 areas, 5 invaded and 3 non-invaded areas (Mucor sp. 1D and Mucor sp. 2I). The remaining 78 morphotypes were from the phylum Ascomycota (96%). The most abundant genus was Fusarium (25%), followed by Aspergillus (12%), Colletotrichum (7.4%), Penicillium (6.2%), and Trichoderma (6.2%).”
- Figure 3) “not the best way to label variants.”
- Line 153) “what does it mean?”
We thank the comment. We suppressed the following part: “. To exclusive morphotypes in each area was 47% of the invasive 153 litter and 53% the from native litter in invaded area, and 50% on each litter in the native 154 area.” And we change the paragraph as follow:
“The distribution of total morphotypes isolates was 54% for the invaded area and 46% for the native area. Evaluating separately each area and each litter studied, in invaded area 25% of morphotypes were isolated from the invasive litter and 29% from native litter, in the native area 23% were isolated from the invasive litter and 23% from native litter.”
- Line 156) “Why do you think that it is the lignocellulotic community?”
We thank the comment. We put an explanation at the beginning of the topic. This expression is used because we choose selective media for isolation.
- Line 181-190) “To answer these questions accurately, it was enough to perform a chemical analysis os the plant mass and determine the ratio of lignin, cellulose, hemicellulose, nitrogen content and an important parameter C:N.”
We thank the comment. The revisor is right, and unfortunately, we don’t have this data. However, we believe that preview studies of Tradescantia fluminensis, a very similar species, provides a good idea about the T. zebrina nutritional content. We change the phrase as follow: “The accumulation of N and other nutrients in the tissues of Tradescantia fluminensis Vell. (Commelinaceae), a morphologically similar species of the same genus and life-form of T. zebrina, accelerates the decomposition of its litter and, together with the large biomass of T. fluminensis, alters the availability of nutrients in invaded forest remnants from New Zealand.”
- Line 192-194) “This could well be estimated, at least approximately.”
We thank the comment. Our native litterbags tried to replicate a natural condition of native litter. So, samples were randomly collected in the forest (close to the litterbag deposition areas) and has no invasive species. The composition of species and types of fragments in the native litter bags was random.
- Line 204-206) “I didn’t understand what you means.”
We thank the comment. We change the phrase as follow: “However, our native litterbags made of a diversity of species and life-forms from the non-invaded areas, are a good approximation of the dynamics of litter decomposition in natural habitats.”
- Line 243 – 4.5 “In vitro experiment) I didn’t see “vitro” in this experiment. This is not an in vitro, but rather a laboratory simulation.”
We thank the comment. In vitro experiment may be used for experiments which are conducted in laboratory conditions, contrary to in situ condition (field experiments).
- Line 419) “In such cases, it is better to use curated databases.”
We thanks the comment. For fungi, the main bibliographies on the subject use these databases. We made a mistake in the part where it says CBS, the correct one is Mycobank. We replace in the text.
Reviewer 2 Report
The title of this manuscript is “The invasive Tradescantia zebrina affects litter decomposition, but it doesn't change the lignocellulolytic fungal community in the Atlantic Forest, Brazil.”. The topic is relevant and presents some interesting findings. I do not have any reservations towards the manuscript in its current form. This article is publishable and should work to clarify the clarify the minor comments below.
1. Line 167-170. The litter of invasive species decomposes quickly, why does it not affect the decomposition rate of litter?
2. Line 171-172. The author believes that the litter decomposition rate of T. zebrina is fast, mainly due to the quality of litter. Therefore, it is recommended to supplement the data about the quality of the litter.
3. Line 323-332. Native species include trees, shrubs, and herbs. Therefore, it is necessary to clarify whether the dry fragments of native species only contain trees, shrubs, or herbs, or all three types of plants. If there are all, how are the species in the dry fragments composition?
Author Response
The title of this manuscript is “The invasive Tradescantia zebrina affects litter decomposition, but it doesn't change the lignocellulolytic fungal community in the Atlantic Forest, Brazil.”. The topic is relevant and presents some interesting findings. I do not have any reservations towards the manuscript in its current form. This article is publishable and should work to clarify the clarify the minor comments below.
- We are pleased that you appreciated our study. We look answer each comment below. We believe our new manuscript version is quite improved.
- Line 167-170. “The litter of invasive species decomposes quickly, why does it not affect the decomposition rate of litter?”
We thank the comment. Although the litter of Tradescantia zebrina decomposes quickly, the decomposition rates of litters (native or invaded) placed at invaded areas are not significantly different when compared with litters placed in non-invaded areas.
- Line 171-172. “The author believes that the litter decomposition rate of T. zebrina is fast, mainly due to the quality of litter. Therefore, it is recommended to supplement the data about the quality of the litter.”
We thanks the comment. Unfortunately, there is no preview study about the T. zebrina litter quality. However, there is studies about Tradescantia fluminensis, a quite similar plant. The next phrase, explains this question: “The accumulation of N and other nutrients in the tissues of Tradescantia fluminensis Vell. (Commelinaceae), a morphologically similar species of the same genus and life-form of T. zebrina, accelerates the decomposition of its litter and, together with the large biomass of T. fluminensis, alters the availability of nutrients in invaded forest remnants from New Zealand [39,40].
- Line 323-332. “Native species include trees, shrubs, and herbs. Therefore, it is necessary to clarify whether the dry fragments of native species only contain trees, shrubs, or herbs, or all three types of plants. If there are all, how are the species in the dry fragments composition?”
We thanks the comment. Sorry, we didn’t understand your question. Dry fragments contain fragments of all native species present in the sample. Our native litterbags tried to replicate a natural condition of native litter. So, samples were randomly collected in the forest (close to the litterbag deposition areas) and has no invasive species. The composition of species and types of fragments in the native litter bags was random.
Round 2
Reviewer 1 Report
Dear Colleagues, The second reading of the manuscript made a much better impression on me. Perhaps the point is not only that the authors made the necessary corrections, but also that I was already more prepared to read this text. I really liked the fact that the authors, despite using not the most fashionable methods, provided a very detailed analysis, convincing statistics and a deep and interesting discussion of the results. This is a good example that it is not always worth striving to use the latest methods of analysis, but it is better to focus on a careful and exhaustive analysis of the available materials. I read the text with great interest, especially since I am currently working with a similar object. In this regard, I would like to invite the authors (if they want, of course) to add one idea, traces of which I also see in the materials with which I work myself:
In lines 27-29, 215-224, 456-457 I would emphasize the idea that in the Atlantic Forests we are dealing not only with a diverse fungal community, but with a stable community, formed over a long time in conditions of high plant diversity. Such a community preserves the historical experience of litter decomposition and easily finds ways to utilize a wide variety of substrates.
I think it is possible that in young, newly established communities, the impact of invasive litter will be much more pronounced. Perhaps for this reason, the results obtained by the authors may not coincide with other published data.
Finally, during the first reading, I did not notice that the authors did not submit the obtained nucleotide sequences to the GenBank. However, I do not think that this can affect the perception of this publication especially since the authors promised to provide data upon request. Just remember to to submit the sequence data you obtained it next time.
Author Response
Dear Colleagues, The second reading of the manuscript made a much better impression on me. Perhaps the point is not only that the authors made the necessary corrections, but also that I was already more prepared to read this text. I really liked the fact that the authors, despite using not the most fashionable methods, provided a very detailed analysis, convincing statistics and a deep and interesting discussion of the results. This is a good example that it is not always worth striving to use the latest methods of analysis, but it is better to focus on a careful and exhaustive analysis of the available materials.
- We are glad about this comment. The challenges of make good science in the third world are huge. As the reviewer mentioned, we choose a way based in a solid experimental design over more “most fashionable methods” (obviously more expensive and out of our support). Furthermore, we were able to include undergraduate students in the isolation processes. In this way, this manuscript becomes a good work not only of research, but also of teaching.
I read the text with great interest, especially since I am currently working with a similar object. In this regard, I would like to invite the authors (if they want, of course) to add one idea, traces of which I also see in the materials with which I work myself:
In lines 27-29, 215-224, 456-457 I would emphasize the idea that in the Atlantic Forests we are dealing not only with a diverse fungal community, but with a stable community, formed over a long time in conditions of high plant diversity. Such a community preserves the historical experience of litter decomposition and easily finds ways to utilize a wide variety of substrates. I think it is possible that in young, newly established communities, the impact of invasive litter will be much more pronounced. Perhaps for this reason, the results obtained by the authors may not coincide with other published data.
- We thank for the precious and precise comment. We change the following phrases:
Lines 27-28: We believe that the high plant richness in the Atlantic Forest enables a highly diversified and stable decomposing biota formed in conditions of high plant diversity.
Lines 220-230: The climate [52], the diversity of decomposing organisms, and the very nature of high diversified plant litter [53], are determining factors in the rate of decomposition [54,55], and regulate the activity of decomposers of organic matter in the soil [56]. The litter heterogeneity of the Atlantic Forest provides a greater diversity of niches for the community of decomposers, keeping the fungal diversity more stable, even under forest gradients [57]. These characteristics favour a highly diversified and active decomposing microbial community [58,59]. The natural range of different litter from the Atlantic Forest provides an opportunity for this decomposing community, formed over a long time in conditions of high plant diversity, which consequently has high taxonomic and functional diversity, to optimise the cycling of nutrients [45,59], including those from invasive litter.
Lines 459-461: The high plant diversity in the Atlantic Forest probably supports a fungal community that preserves the historical experience of a wide variety of litter, and easily finds ways to utilize different substrates, including exotic ones.
Line 476-479: We thank the Graduate Program in Neotropical Biodiversity at UNILA for support and structures; Silvane Gonçalves Santos for her excellent help in making the litter bags; the reviewers for the excellent comments and suggestions that improved a lot our manuscript.
Finally, during the first reading, I did not notice that the authors did not submit the obtained nucleotide sequences to the GenBank. However, I do not think that this can affect the perception of this publication especially since the authors promised to provide data upon request. Just remember to to submit the sequence data you obtained it next time.
R. The reviewer is right. We will provide the nucleotide sequences to the GenBank. If necessary, we can provide the full data anytime.